**Data Availability Statement:** All relevant data (several excerpts from focus groups) are within the paper. Verbatim transcriptions cannot be shared

# Interdisciplinary stratified care for low back pain: A qualitative study on the acceptability, potential facilitators and barriers to implementation

**Carmen Caeiro**[1]*, **Helena Canhão**[2], **Sofia Paiva**[1], **Luís A. Gomes**[1], **Rita Fernandes**[1], **Ana Maria Rodrigues**[2], **Rute Sousa**[2], **Fernando Pimentel-Santos**[2], **Jaime Branco**[2], **Ana Cristina Fryxell**[3], **Lília Vicente**[3], **Eduardo B. Cruz**[1]

1 Physiotherapy Department, Setúbal Polytechnic Institute, Setúbal, Portugal, 2 EpiDoC Unit, CEDOC, NOVA Medical School, Lisbon, Portugal, 3 ACES Arrábida, The Regional Health Administration of Lisbon and Tagus Valley, Setúbal, Portugal

* carmen.caeiro@ess.ips.pt

## Abstract

### Background and objective

The SPLIT project aims to introduce an interdisciplinary stratified model of care for patients with low back pain. This study aimed to explore the acceptability and identify potential barriers and facilitators regarding the upcoming implementation of this project, based on the general practitioners' and physiotherapists' perceptions.

### Methods

A qualitative study was carried out supported by two focus groups, which were conducted by two researchers. A focus group was carried out with each professional group. One focus group included six general practitioners and the other included six physiotherapists. The focus groups were based on a semi-structured interview schedule, audio-recorded and transcribed verbatim. A thematic analysis was conducted.

### Results

The participants explored aspects related to the acceptability of the SPLIT project, emphasising the satisfactory amount of effort that is expected to be required for its implementation. Potential facilitators to the implementation of the model were identified, such as the participants' motivation. Potential barriers were also explored, with particular emphasis on the challenges related to the change of routine care. Lastly, the need for particular adjustments in the health services was also highlighted.

### Conclusions

This study's participants highlighted the feasibility and acceptability of the SPLIT project. The identification of potential barriers and facilitators to its implementation also attained

publicly due to ethical reasons. Transcripts are available upon reasonable request to Specialized Ethics Committee from the School of Health Care (ceei.ctc@ess.ips.pt).

**Funding:** The work presented in this paper is associated to a research project co-financed by the LISBOA 2020 – Regional Operational Program for Lisbon and Vale do Tejo and the Portuguese Science & Technology Foundation (Grant#SAICT-POL/23439/2016|LISBOA-01-0145-FEDER-023439).

**Competing interests:** The authors have declared that no competing interests exist.

major relevance to better prepare the upcoming implementation of this project. The generalizability of findings to the larger population of relevant practitioners is limited, since only two focus groups were carried out. Therefore, this study's findings should be considered in terms of transferability to contexts that may have some similarities to the context where the study was carried out.

## Introduction

Low back pain (LBP) has prevailed worldwide as one of the three leading causes of non-fatal health loss for almost three decades [1]. It has been identified as a leading cause of years lived with disability and its burden has steadily increased, despite the extended research and health care resources that have been developed in this area [2–4]. In Portugal, LBP is the most prevalent musculoskeletal condition, with a significant proportion of patients reporting disability as well as a high level consumption of health services [5–7].

It has been suggested that the current clinical practice has not followed clinical guideline recommendations and may not be delivering the best treatment to Portuguese patients with LBP [8]. In particular, the advice to rest, analgesic, anti-inflammatories and medical imaging prescriptions as well as sickness certifications were amongst the most common approaches to Portuguese patients with LBP in primary care [8]. These findings were consistent with those from worldwide studies, which have called attention to the poor outcomes of this approach on pain, disability and sense of well-being in patients with LBP [9–13].

Research has recommended a greater use of biopsychosocial model in the management of patients with LBP [3,14,15]. Stratified care for LBP has tried to reassure the aforementioned model in clinical practice, by targeting treatment to subgroups of patients based on their key characteristics such as prognostic factors, likely response to treatment and underlying mechanisms [16]. The STarT Back—Subgroups for Targeted Treatment is a stratified model of care that has been developed and tested by researchers from Keele University over the last 15 years [17]. According to this model, patients with LBP are assessed with the Keele STarT Back Tool (KSBT) that categorizes them into three subgroups based on their risk (low, medium or high) of developing persistent disabling pain. Based on this categorization, patients are provided with matched treatments [18]. Research focused on the STarT Back has demonstrated better clinical outcomes at 4 and 12 months, as well as lower health costs when compared with current best practice [19]. Further research on the STarT Back has strengthened its clinical and cost effectiveness and demonstrated its feasibility in primary care in the UK [20,21]. Additionally, one study has reported positive results in Ireland [22]. However, further research is needed to determine if these benefits are still evident when compared to current practice in other countries, and six studies are in progress [17].

Considering previous findings from research on the STarT Back, the SPLIT project aimed to introduce a similar approach, involving General Practitioners (GPs) and Physiotherapists (PTs) in the triage and targeted treatment for patients with LBP, in a Group of Portuguese Health Centres (ACES Arrábida), in primary care. The KSBT was translated into Portuguese and validated in Portuguese population [23], and then funding was obtained for the SPLIT Project in 2018.

Since previous qualitative research has called attention for potential barriers that may prevent the successful implementation of the STarT Back in the UK as well as in other countries [24–27], the SPLIT research team has introduced several adjustments to the original model

since its early conception. For example, rheumatologists and PTs from the research team delivered a training program, which was tailored to each professional group. GPs attended two 2-hours workshops focused on the triage process as well as on the model of care proposed for patients with LBP, inspired on the STarT Back. PTs also attended the latter workshop, as well as to a 4-days course focused on the assessment and treatment of patients classified, by the KSBT, with low, medium and high risk of developing persistent disabling pain. A book in Portuguese including detailed information about the contents of the course was also developed in order to support the PTs while attending the course and afterwards in their clinical settings.

In order to promote the successful implementation of the SPLIT project, the exploration of the perceptions of the GPs and PTs, who had attended the aforementioned training, regarding its upcoming implementation was considered fundamental. In particular, the knowledge about acceptability as well as potential facilitators and barriers prior to its implementation was considered essential to better prepare researchers and health professionals.

## Materials and methods

### Study design

In order to gain insights into the GPs and PTs perceptions regarding the upcoming implementation of the SPLIT project, a qualitative study was considered most suitable. The criteria for reporting qualitative research (COREQ) provided important guidelines for this study design as well as data analysis [28]. In particular, a Thematic Analysis (TA) was carried out to identify, analyse and report common themes and sub-themes within and across data [29]. Some authors have characterized TA as a tool to use across different methods, however, others argued that it might be seen as a method in its own right [29,30]. In this study, TA was used as a methodological approach, particularly focused on a rich description of data, with less emphasis on producing an in-depth interpretation that would be expected from other methodological approaches, such as grounded theory or interpretative phenomenological analysis [31,32].

Ethical approval was granted by the Specialized Ethics Committee from the School of Health Care, Setúbal Polytechnic Institute (36/LN/2018).

### Context

The Portuguese health care system is organized around an NHS model, whose planning and regulation is developed by the Ministry of Health, with some increasing responsibilities delegated to regional bodies [33]. All residents have access to the health care provided by the NHS and participate in health care financing via co-payments and co-insurance [34–37]. The primary care follows the organization of the NHS, including regional administration bodies that manage groups of primary care centres. The SPLIT project was planned in partnership with one of the aforementioned administration bodies (ARS Lisbon and Tagus Valley) to be implemented in 1 Group of Health Centers (ACES Arrábida). This group of health centers provides health services to a population of 233.516 individuals, living in the municipalities of Palmela, Setúbal and Sesimbra [38]. From the 22 health units included in this group of health centers, 7 were selected for piloting the project, based on the proportion of geographical area covered by ACES Arrábida (1 health unit from Palmela, 2 health units from Sesimbra and 4 health units in Setúbal). A total of 52 GPs and 3 PTs works in these 7 health units [38].

Information on the typical journey of a patient with LBP in primary care in Portugal is quite scarce. Recent literature has highlighted the patients'multiple journeys in the Portuguese NHS [39]. The vast majority of individuals start by attending an appointment with their GP, who make the decision about the need for referral to another health professional [40]. Patients

consider the access to health care and the process of diagnosing and referring as difficult and prolonged [40] and tend to go to NHS and private practice interchangeably [39].

Despite the limited information on the specific waiting time of patients with LBP in the health units where the SPLIT project was going to be implemented, it was known that the process was lengthy, as there were no physiotherapy services available for these patients and they needed referral for outsourced physiotherapy services.

## Participants

Since the SPLIT project was routed in an interdisciplinary work involving GPs and PTs, the collection of data regarding the perceptions of both professionals groups was considered fundamental. Accordingly, twelve participants (six GPs and six PTs) were recruited from those who had completed the training program that had been delivered. As there were only 3 PTs working in the health units where the SPLIT project was going to be implemented, the participation of more PTs from similar units was considered. The attendance to all sessions of the program was the only inclusion criteria for the participation in this study. The potential participants were randomly selected and invited to participate by e-mail, two weeks prior to the focus group. The e-mail included an invitation with a detailed explanation of the study, a consent form as well as the interview schedule. The potential participants were given the opportunity to consider if they wished to participate and, in case of a positive decision, were asked to reply to the e-mail in one week. All participants, who were invited to participate, accepted to take part.

## Data collection

Two focus groups (one with GPs and one with PTs) were carried out in order to collect data. The focus group is defined as a discussion-based interview that produces data and insights via group interaction, which would be less accessible without the interaction founded in a group [41,42]. The focus groups were selected as a method for data collection since the aim of this study was focused on understanding the combined local perspectives of each professional group participating in the implementation of the SPLIT project. The research team aimed to explore potential agreements or disagreements as the discussion progressed and to grasp each professional group perceptions regarding the future implementation of the SPLIT project. The focus group sessions had a 60-minute duration and were held by the first author (CC), while SP took notes in order to summarize the main topics discussed during the session. CC had previous experience in facilitating focus group discussions and SP had experience as a research assistant in focus groups.

Both focus groups were carried out immediately after the end of the previously mentioned training program. The five stages for moderating focus groups suggested by Finch et al (2014) were followed: scene setting and ground rules; individual introduction; the opening topic; discussion; and, ending the discussion [42]. The discussion was based on a semi-structured interview schedule based on exploratory open-ended questions. Particularly, open, transitional, key questions and probes were used, following Krueger and Casey's recommendations [43]. The questions were grouped into two sections, as outlined in Table 1. The first section included generic questions, designed to explore the participants'perceptions regarding the acceptability of the SPLIT project; the second section contained specific questions designed to explore the potential facilitators and barriers to the upcoming implementation of this project. Since questions on the first section were generic, they were asked to both professional groups. The second section was tailored to each professional group, according to the expected role on the SPLIT project.

**Table 1. Interview schedule.**

| |
| --- |
| **Section I. GPs and PTs'overall perceptions regarding the acceptability of the SPLIT project** |
| 1. What are your perceptions regarding the acceptability of the stratified model of care included in the SPLIT project? |
| • What are your perceptions regarding the use of the KSBT? Is it acceptable to you making decisions regarding patients' treatment according to the outcomes of this tool? |
| 2. Could you please compare this model with your current practice in patients with LBP? |
| • Do you think this model add any value to your practice? |
| 3. In your opinion, what may be the impact of the implementation of the SPLIT project on the approach to patients with LBP? |
| • Do you think it may impact on: your work volume?; your relationship with other professionals?; patients' access to health care?; patients' outcomes?; any other aspects in the organization of the health unit where you work? |
| 4. Could you please identify any potential barriers to the implementation of the SPLIT project in the health unit where you work? |
| 5. Could you please identify any potential facilitators to the implementation of the SPLIT project in the health unit where you work? |
| **Section II a. GPs'perceptions regarding the upcoming implementation of the SPLIT project** |
| **1.** What would help you integrating the KSBT in your practice? |
| 2. In your opinion, additional training should be delivered in order to promote the development of competences to implement the stratified model of care included in the SPLIT project? |
| • If yes, what particular contents? |
| 3. Are there any important resources for the successful implementation of this model missing in the health unit where you work? |
| • If yes, what particular contents? |
| **Section II b. PTs'perceptions regarding the upcoming implementation of the SPLIT project** |
| **1**. The implementation of one single session for patients classified with low risk of developing persistent disabling pain is one of the recommendations of this stratified model of care–what do you think about this? Is it too different from what you have been doing? |
| • Do you expect any difficulties following this recommendation? |
| • What would be important to help you following this recommendation? |
| 2. The implementation of *hands-off* interventions, mainly education and exercise, for patients classified with high risk is a key point in this model—What do you think about this? Is it too different from what you have been doing? |
| • Do you expect any difficulties following this recommendation? |
| • What would be important to help you following this recommendation? |
| 3. What about the recommendations for the treatment of patients classified with medium risk? |
| 4. In your opinion, additional training should be delivered in order to promote the development of competences to implement the stratified model of care included in the SPLIT project? |
| • If yes, which? |
| 5. Are there any important resources for the successful implementation of this model missing in the health units from the group of health centres where you work? |
| • If yes, which? |

The focus group sessions were audio-recorded and transcribed verbatim by SP. The participants'names were replaced by the pseudonymous they had selected prior the focus group session.

## Data analysis

Data analysis followed the six-phase process suggested by Braun and Clarke (2006). The first phase involved the researchers'familiarization with data by reading and re-reading the transcripts and listening to the recordings, while taking initial notes about the data and potential topics for further exploration. The following phase included a systematic coding of the data. The codification was guided by this study's aims, meaning that the researchers were looking

for information related to the acceptability, facilitators and barriers to the upcoming implementation of the SPLIT project. Nevertheless, it was agreed that the codification should remain opened in order to allow the inclusion of any other relevant topics that emerged from the data. Two researchers (CC and SP) independently coded the transcripts and produced a list of codes and the relevant excerpts to each code. Each excerpt was identified by a sequence of words and numbers (professional group, pseudonymous, page number of the transcript, line number) in order to allow an easy identification in the transcript. The researchers discussed the codes and examined their scope and relevance. This discussion prompted the beginning of the third phase of analysis, which was focused on the development of initial themes. At this phase, the codes were examined in more depth in order to identify similarities and differences and then were clustered to form the first sub-themes. The connections among the sub-themes were discussed in order to develop potential themes. A "thematic map" of the analysis was developed as an outcome of this discussion. In the following phase, CC reviewed the previously identified themes and sub-themes against the data in order to check that the themes together addressed the study's aims in a meaningful way, the most relevant data were captured and no relevant information was discharged. Themes and sub-themes were refined and brought to discussion with SP and EBC, who agreed with the proposed themes and sub-themes. This prompted the beginning of the fifth and sixth phases that were focused on naming the final themes and sub-themes and writing-up, respectively. In fact, both phases overlapped since writing-up involved further analysis and subsequent reorganization of themes, sub-themes and relevant excerpts [29,30].

## Results

Twelve participants (9 female, 3 male) were included. Five of the participants were 30–39, four were 40–49, and three were 20–29, 50–59 and 60–69, respectively. The GPs'group had an average of 13.17 years of professional experience, from which 10.17 years were based in primary care. The PTs'had an average of 21 years of professional experience, from which 13.87 years were based in primary care.

Four themes and twelve sub-themes emerged from data analysis, as outlined in Table 2. These themes explored the aspects that this study's participants considered relevant to

**Table 2. Main themes and sub-themes emerged from data analysis.**

| Themes | Sub-themes |
|---|---|
| **1. Acceptability of the SPLIT project** | **1.1.** The SPLIT project feasibility |
| | **1.2.** The potential increase of the quality of health care provided |
| | **1.3.** The potential value creation and innovation |
| **2. Facilitators to the implementation of the SPLIT project** | **2.1.** The motivation to improve the health care provided |
| | **2.2.** The coherence between the SPLIT project and the aims and mission of primary care |
| | **2.3.** Interdisciplinary work and centralization of care in the health units |
| **3. Barriers to the implementation of the SPLIT project** | **3.1.** The change of routine care |
| | **3.2.** The patients' resistance to a different treatment approach |
| | **3.3.** The management of rooms in health units |
| **4. Tailored planning and specific adaptation to the context** | **4.1.** The health professionals'need for specific training |
| | **4.2.** The reorganization of the health services |
| | **4.3.** The development of new strategies to facilitate communication between GPs and PTs |

promote the effective and successful implementation of the SPLIT project in the context of primary care. Excerpts from the focus groups are presented to support the analysis.

### Theme 1: Acceptability of the SPLIT project

In the first theme, the participants'perceptions regarding the acceptability of the SPLIT project were examined. This theme included three sub-themes that explored the main factors that justify the participants'perception of acceptability: the feasibility; the potential increase of the quality of health care provided; and, the potential value creation and innovation.

**Sub-theme 1.1. The SPLIT project feasibility.**   From the GPs'point of view the acceptability was mainly related to the reduced time required from their appointments, since they considered being very restricted in what concerns to the time available to attend patients. The referral to physiotherapy in the context of this project was perceived as both acceptable and advantageous for two main reasons: quickness, as GPs do not need to fulfil the KSBT prior to referral; and, lowered costs since it allows the referral for physiotherapy in the health unit, as opposed to the referral to outsourced physiotherapy service that happened previously.

> " . . . the GPs won't have extra work (. . .) We'll consider the need for referral to physiotherapy. . . But, we won't have to do anything extra in the few minutes we have to pay attention to each patient." (GP Maria, 15, 673)

> "If this tool [KSBT] were introduced in the appointments with GPs it wouldn't be possibly used. So, it is very important to keep it with the PTs, as this facilitates the referral." (GP João, 5, 208)

> "On the other hand, we should consider the potential costs reduction. The costs associated to the referral to physiotherapy inside the health unit will be lower than those from the outsourced physiotherapy services. So, this is an important factor for the acceptability and adherence to this project." (GP João, 2, 49)

Both professional groups valued the use of the KSBT and considered it as an important instrument for triage and stratification of patients according to their risk of developing persistent disabling low back pain. In particular, the PTs, who were expected to use this tool to assess patients, emphasised the importance of its easy usage as well as the usefulness of the information obtained from it for their clinical reasoning.

> "This tool is feasible in our context, because it includes only nine questions, which are easy to answer (. . .). The outcome helps us on stratifying the patient and identifying treatment priorities." (PT Eunice, 5, 198)

> "So, this tool is going to be very useful because it's going to guide our practice, by stratifying patients. . . This is going to help us in clinical reasoning and in the decisions regarding treatment." (PT Sofia , 4, 154)

**Sub-theme 1.2. The potential increase of the quality of health care provided.**   Both professional groups emphasised the potential increase of the quality of health care provided, when they compared the SPLIT project to the care that was being provided at the time of the focus groups. This potential improvement was justified by three factors: firstly, the expected reduction of waiting time to start physiotherapy; secondly, the expected provision of care that is both evidence-based and personalised according to the patient's risk of developing persistent

disabling pain; and, thirdly, the expected reduction of the number of sessions needed for discharge.

> "Ok, currently I assess a patient and identify that he needs physiotherapy. But, when does he start doing physiotherapy? In a month? Two months? He waits for another appointment with a physician in the outsourced physiotherapy service and then he waits again to start doing physiotherapy sessions. . . It's too much time (. . .) I think this is meaningful for the patient. . ." (GP Lita, 7, 300)

> "This model helps us personalizing treatment according to the patient's sub-group (. . .) and, thus, providing a more effective and evidence-based response." (PT Cristina, 3, 104)

> "We [GPs] are aware some patients do physiotherapy in an outsourced service and have some outcomes after a number of sessions, but other patients in another service need much more sessions to obtain those outcomes." (GP Sophie, 7, 325)

**Sub-theme 1.3. The potential value creation and innovation.** While considering the acceptability of the SPLIT project, this study's participants also emphasised the potential innovation and value added to the services provided by health units from primary care. The aforementioned innovation and value added were explained by the implementation of an interdisciplinary stratified model of care that responds to LBP patients'needs and by a more effective management of time and human resources.

> "It [the SPLIT project] will add a new offer to the list of health services provided by this group of health centres. . . Something that it's new and different. . . And, taking into account the scientific evidence presented by the research team, it seems better than the current clinical practice. . ." (GP João, 1, 46)

> "To be able to better manage human resources is an added value to our employee. . . PTs are scarce in primary care (. . .) So, this may help moving PTs from practices that are not evidence-based towards new and more effective treatments." (PT Guilherme, 3, 83)

## Theme 2. Facilitators to the implementation of the SPLIT project

In the second theme, the potential facilitators to the implementation of this project were identified. Three main facilitators were explored: the participants'motivation to improve the health care provided; the coherence between this project and the aims and mission of health units from primary care where it was going to be implemented; and, the promotion of interdisciplinary work as well as the centralization of care in each unit.

**Sub-theme 2.1. The motivation to improve the health care provided.** Both groups considered their motivation to implement the SPLIT as an important facilitator. This motivation was grounded on the expectation of increasing the quality of health care provided. Nevertheless, the PTs called attention to the expected challenges and difficulties associated with the need to change their practice. Additionally, the GPs highlighted the importance of having regular contact with elements from the research team in order to maintain their motivation.

> ". . . this is a facilitator. . . having the guarantee that I'm doing evidence-based treatment and I'm contributing to improve patients' health condition." (PT Maria, 4, 146)

> "This is going to be the biggest challenge: to change what we used to do and then assess the outcomes from an approach that is more focused on patient education and exercise" (PT Sofia, 5, 192)

**Sub-theme 2.2. The coherence between the SPLIT project and the aims and mission of primary care.** According to this study's participants the alignment between the SPLIT project and the mission of health units from primary care was considered one of the main facilitators to its implementation. On the one hand, the SPLIT project was expected to implement a model of care that may help to reduce the substantial impact of LBP on both patients and the NHS. On the other hand, it may contribute to achieve the goals planned by the health units from primary care. More specifically, the GPs and PTs called attention to the potential improvement on health indicators for medical prescription, with a potential reduction on medical prescription of medication and medical exams, such as X-rays.

"Without this stratification we would probably use the same treatment for everyone. Some would become better, some wouldn't. We would have more costs because we would probably prescribe more unnecessary medical exams . . ." (GP Sophie, 6, 268)

"In LBP, the prescription of anti-inflammatory and medical exams needs to be reduced. This [the SPLIT project] may help patients but also health units, which have health indicators to achieve. . . So, for me, this is one of the major facilitators." (PT Guilherme, 17, 665)

**Sub-theme 2.3. Interdisciplinary work and centralization of care in the health units.** The SPLIT project alignment with the multidisciplinary nature of services provided by the health units from primary care and the simultaneous maintenance of the autonomy of each professional group are considered key facilitators. The GPs considered the SPLIT project might promote interdisciplinary health care, since it would possibly contribute to the increase of knowledge of the specific contribution of each professional group in the approach to LBP patients. Additionally, the centralization of care in the health units from primary care, without the need for referral to outsourced physiotherapy services, was also identified as an important facilitator, since it was expected to facilitate the patients' access to health care.

" (. . .) So. we'll have physios working directly with us. . . (. . .) So, we know what they have to offer, what may be the advantageous for patients. And, this makes us feel more confident to recommend physiotherapy" (GP Sophie, 7, 329)

". . . I think this [the SPLIT project] may prompt and be a good example of interdisciplinary care. . . even for projects in other health problems (. . .) it puts patients in the centre, may increase efficacy and reduce costs." (GP João, 7, 316)

## Theme 3: Barriers to the implementation of the SPLIT project

In the third theme, the barriers were discussed, with particular emphasis on three potential issues: the change of routine care; the patients' resistance to a different treatment approach; and, the management of rooms in health units.

**Sub-theme 3.1. The change of routine care.** Although the personal motivation to implement the SPLIT project was considered as a facilitator, the participants also stressed that the change of routine care might be a barrier. The GPs identified two main problems related to the referral to physiotherapy: "reduced" referrals—due to reduced time of appointments, GPs were afraid of not referring patients who would benefit from physiotherapy; and, "incorrect" referrals—GPs anticipated the possibility of referring patients with exclusion criteria.

". . . some days are chaotic. . . sometimes our levels of stress are at the top. . . we don't have time even for the basic procedures . . . let alone the new procedures that constitute an extra,

of course it's a nice extra, but they will require extra time since we're not used to them. . ." (GP Diana, 18, 821)

" . . . the reduced referral may be an issue, and also an incorrect referral, I mean we may not consider the inclusion and exclusion criteria for patients to be included in the SPLIT project" (GP João , 10, 433)

The PTs also emphasised the need to change their clinical practice, mainly with patients classified with high risk, as a potential issue. This professional group anticipated difficulty in implementing hands-off treatments focused on patient education about pain, as this would be very different from what they were used to do.

"Well I can learn the neurophysiology of pain. . . it's all about reading the references. . . But, the cognitive-behavioural model for patient education. . . For me, this is definitely the most difficult." (PT Guilherme, 18, 702)

**Sub-theme 3.2. The patients' resistance to a different treatment approach.** Both groups anticipated the potential resistance from patients to a different clinical approach as a potential barrier. However, they had different perceptions regarding the group of patients that might be more resistant–the GPs considered the patients classified with low risk, while the PTs indicated the patients classified with high risk.

" . . . just one session? We know patients and we know how difficult it is. . . Are they going to be ok with this idea? I ask myself. I'm not sure." (GP Lita, 5, 190)

" (. . .) I don't think the low risk group is going to feel the need for more sessions because the prognosis is quite positive (. . .) But, the patients classified with high risk. . . we'll need to manage this approach with caution because it's a completely different treatment [from what they were used to]. . ." (PT Sofia, 5, 183)

Despite the aforementioned resistance, the GPs emphasised patients' lack of satisfaction with the outcomes they have achieved with previous health care and considered this could reduce resistance to new treatments.

" (. . .) they [patients with LBP] want new solutions. They want different treatments. . . And, it's important to explain that having an X-ray won't release pain. . . (. . .) they need a specialized treatment. . ." (GP Alex, 5, 196)

**Sub-theme 3.3. The management of rooms in health units.** The management of material resources was also identified as a potential barrier for the implementation of the SPLIT project. In particular, the participants identified the lack of rooms with adequate dimensions to carry out physiotherapy group sessions. Nevertheless, both groups seemed to be motivated to look for solutions for this problem.

"We are very limited in what concerns rooms. . . (. . .) Of course, with good will we'll find a suitable place, but it's very difficult to manage this issue." (GP Lita, 9, 378)

"For example, we just have one room for group sessions in the health unit where I'm working. . . (. . .) This room is used for different purposes, by several health professionals, so we have to manage this very carefully and book our sessions on the available time schedule" (PT Ana, 14, 572)

### Theme 4 –Tailored planning and specific adaptation to the context

In the fourth theme, the need for a tailored planning and specific adaptation to the contexts where the SPLIT project was going to be implemented was emphasised. Three main aspects were explored: the health professionals'need for specific training; the reorganization of the services; and, the development of new strategies to facilitate communication between GPs and PTs.

**Sub-theme 4.1. The health professionals'need for specific training.** Both professional groups emphasised the importance of specific training included in the context of the implementation of the SPLIT project. The PTs explored the role of this training in more depth and considered it as fundamental for what they called *"a paradigm shift"*. This shift seemed to be related to the focus on promoting patients' self-management, rather than on the identification and treatment of the body structure in the origin of pain. According to the PTs the training provided was considered adequate for starting the implementation of this project. However, they identified two areas (patient education and exercise) in which they perceived to be less competent and highlighted them as learning needs for future.

> "In my opinion the training covered all important aspects (. . .) It has been a good starting point. . ." (PT Guilherme, 3, 93)

> "I need to learn more about patient education, mainly how to use strategies from the cognitive-behavioural model in order to carry out education about the neurophysiology of pain" (PT Guilherme, 17, 678).

> "Also, from our graduation. . . We have little knowledge about exercise prescription. . ." (PT Sofia, 18, 696)

**Sub-theme 4.2. The reorganization of the health services.** Both groups identified the need to reorganize health services in each unit in order to promote the successful implementation of the SPLIT project. The need for creating a specific period in the PTs'agendas focused on treating patients with LBP was highlighted. This was identified as a priority, since PTs working in the health units where the SPLIT project was going to be implemented did not use to treat patients with LBP (these patients used to be referred to outsourced physiotherapy services).

> "To have a physiotherapist available twice a week to receive the patients that we assessed is fundamental. . . Also, the administrative assistants should be involved, they should know the schedule and inform patients. . . And we [GPs] should be able to inform patients: on that day, at that time, you [the patient] can come to the health unit and speak with the physiotherapist. I think this is very important" (GP Diana, 13, 602)

> " . . . we need to create a specific period on our schedules to treat these patients" (PT Ana, 10, 409)

**Sub-theme 4.3. The development of new strategies to facilitate communication between GPs and PTs.** Finally, the need for new strategies that may facilitate communication between GPs and PTs was highlighted. This sub-theme was explored in more depth by the GPs. Since the referral had been previously identified as a potential barrier, the GPs suggested some strategies to promote the integration of referral of patients with LBP to physiotherapy into their routine care.

"We have so much work to do each day. . . frequently, our IT has problems and this steals time from the appointment. . . We have just 20 minutes to that patient. . . if we don't have a reminder we will forget to do this referral. . ." (GP Maria, 16, 721)

" . . . I suggest something simple, for example a stick with the word SPLIT. . . Something that we can put in our desk, on our computer. . . This is enough to remind us" (GP João, 17, 754)

Additionally, they emphasised the need for feedback from PTs regarding two aspects: the referral, in what concerns to its agreement with the inclusion and exclusion criteria for the SPLIT project, particularly, in the first months of implementation; and, the outcomes achieved with each patient referred. In order to facilitate this process, regular meetings with both professionals groups were proposed.

"This feedback may happen on a monthly basis, or each three months, whatever. . . But, it should be incorporated into the GPs meetings. . . (. . .) Feedback about the referrals themselves, and also about the outcomes achieved with patients. . . this is quite important." (GP João, 19, 852)

"The physiotherapist is there so. . . we could increase our communication. . . We [GPs] have regular meetings, and it may be possible for her [PT] to come. . . (. . .) Or, she may think about the adequate frequency. . ." (GP Lita, 17, 787)

## Discussion

The SPLIT project aims to improve clinical outcomes and cost-efficiency of services delivered to patients with LBP as well as improve their experiences in the NHS. Given the GPs and PTs involvement in all changes proposed to care delivery, it became important to explore their perceptions regarding the acceptability, potential facilitators and barriers before the implementation of the aforementioned project. Qualitative research methods have been considered useful for the development, evaluation and implementation of complex interventions in health care [44]. Previous qualitative studies have already been published on the perceptions of GPs as well as PTs about this topic [24–27], however, data from both groups have been explored separately. To the authors'knowledge, this is the first study encompassing and combining the perceptions of both professional groups, which were expected to implement an interdisciplinary approach. The exploration of the potential barriers and facilitators to the implementation of an intervention, rather than simply focusing on observations before and after its implementation may provide an opportunity for future triangulation of data. This could increase the understanding of both the outcomes of the project as well as the implementation process and how the latter influences the health care delivery and its outcomes [45].

This study's participants highlighted the feasibility of the SPLIT project. On the one hand, the GPs valued the possibility of referring patients with LBP for physiotherapy in the health unit without adding extra work and consuming time from their appointments. On the other hand, the PTs emphasised the easy usage of KSBT as well as the usefulness of the information obtained from it for their clinical reasoning. The use of KSBT by PTs, rather than by GPs was introduced in the SPLIT project, since previous research carried out in the UK has suggested GPs'reluctance in using the tool due to time constraints and pressures inherent to a busy practice [24]. On the other hand, the study by Karstens and colleagues (2015) has called attention to the lack of agreement among German GPs regarding the most appropriate KSBT administering health professional. Findings from the current study as well as the latter two studies

called attention to differences in the organization of health care services and health professionals' perceptions regarding the use of KSBT across different countries. Despite the demonstration of STarT Back clinical and cost effectiveness in the UK [19], further evidence for the adaptations required to test and implement it in other countries is still needed [17]. In the current study, both professional groups anticipated a potential increase of the quality of health care provided, following the introduction of the SPLIT project in their health units. This was related to a potential reduction of waiting time to start physiotherapy as well as to the provision of evidence-based and patient-centred care. The future implementation of the SPLIT project was viewed as an important step to improve the experiences of patients with LBP, by providing a new service designed to enable a better management of time and human resources and, simultaneously, respond to patients' needs. This study's participants'perceptions regarding the potential impact of the SPLIT project were aligned with the advantages associated with the implementation of models of care that have been reported in previous research [46,47]. Moreover, both professionals'groups demonstrated awareness of some inconsistencies between their current clinical practice and the international guidelines for treating patients with LBP. This awareness seemed to promote the acceptability of the SPLIT project. Acceptability is a multi-faceted construct that reveals the extent to which people delivering or receiving a health care intervention consider it to be appropriate, taking into account anticipated or experiential cognitive and emotional responses to the intervention [48]. It has been considered a key element in the design, evaluation and implementation of health care interventions [48]. Although the interview schedule used in this study was not designed to explore all dimensions comprised in acceptability, the participants discussed several of them such as their affective attitude about the project, the perceived amount of work required, the extent to which the intervention was perceived as likely achieve its purpose, as well as the extent to which they understood the project and how it would work [48]. The acceptability demonstrated by this study's participants could be viewed as an important indicator that the implementation of the SPLIT project will be possibly delivered as intended by the research team [49].

GPs and PTs, who participated in this study, perceived their motivation to take part in the implementation of the SPLIT project as a facilitator. The importance of health professionals'engagement in the operating of workforce changes had already been stressed in previous research [45]. Furthermore, this study's participants also considered the promotion of interdisciplinary work as an important facilitator. This finding is coherent with previous studies that had already called attention to the need to improve inter-professional collaboration to successfully implement the STarT Back [27]. While in previous research GPs'limited levels of trust in PTs'work and education was identified as a significant challenge [27], in the current study GPs were the professional group who stressed this cooperation the most and emphasised the importance of knowing in more depth the work of PTs. Additionally, the centralization of care in each unit without the need for referral to outsourced physiotherapy services was considered one of the most important facilitators, since this is expected to increase patients' accessibility to health services and reduce the costs for the NHS.

This study's participants also identified potential barriers to the implementation of the SPLIT project. The change of routine care was stressed by both professionals groups and had already been identified in previous research [24,25,45]. The GPs'anticipated struggle to remember identifying patients who might benefit from the project had already been reported by GPs, who had participated in the STarT Back Trial, in the UK. Similarly, the PTs, who participated in this study, highlighted the challenges inherent to the change of their treatment from a biomechanical perspective to a biopsychosocial model, as PTs in the UK had identified before [25]. Furthermore, patients' resistance to a different treatment approach was also identified as a potential barrier, although GPs and PTs were not in agreement regarding the group of

patients that might be the most reluctant. Patients' resistance was not particularly emphasised in former studies [24,26,27], with the exception of the study by Sanders and colleagues (2013) that mentioned the PTs'anticipation of patients' resistance regarding the discussion of psychosocial dimension of LBP. Previous research has focused the resistance to new health care interventions on health professionals, due to the perceived threat to their autonomy [24]. The difference found in the current study may be related to the aforementioned GPs and PTs'awareness of some inconsistencies between their current clinical practice and the international guidelines for treating patients with LBP. Nevertheless, it is important to take into consideration the potential impact of social desirability on the participants'interventions in the focus groups, which were conducted by two members of the research team implementing the SPLIT project. Moreover, the lack of available appropriate rooms was also identified as a potential barrier. This seemed to be a contextual issue, which might be related to the recent financial crisis that has affected the quality of health care provided in Portugal [50]. In spite of this potential problem, both professionals groups seemed to be committed to find solutions.

Finally, both GPs and PTs that participated in this study stressed the need for a tailored planning and specific adaptation of the project to the context where it was going to be implemented. Both professionals groups valued the training delivered and considered it adequate to prompt the implementation of the SPLIT project. Indeed, this training program integrated specific adaptations to the Portuguese context. For example, workshops were used for answering questions from GPs and PTs about their role on this process and two dates were offered for each workshop in order to facilitate adherence. A specific program was designed for PTs, including seminars, practical sessions, role-plays and mentoring sessions in the PTs'clinical settings were also planned to be operating on a monthly basis in the first six months of implementation. Additionally, two members from the research team were also available to provide assistance by telephone during this initial stage of implementation. Feedback about referrals and clinical outcomes was also planned to be delivered to GPs and PTs on a regular basis, in the first six months of implementation. Although the practicalities of conducting the SPLIT trial had already been considered by the research team, the GPs and PTs emphasised new aspects in the focus groups that deserved further attention. For instance, the need for reorganizing the health services was stressed by this study's participants. This seemed to demonstrate their willingness regarding the implementation of a model of care for patients with LBP in primary care. Additionally, both professional groups, in particular GPs, emphasised the need for new strategies to facilitate inter-professional communication and work team. These findings indicated that there might be a good adherence from GPs and PTs to this project, although it is known that the adoption rates for complex interventions tend to be low, which is partially explained by poor or ineffective implementation [51]. Further research on GPs and PTs perceptions after an initial period of implementation is recommended and may provide important knowledge, especially if combined with the analysis of clinical outcomes and cost-efficacy. Furthermore, it is important to assess acceptability and clinical outcomes from the perspective of patients with LBP, since it may offer a deep understanding of the adhesion to the interventions proposed [48,52].

Some limitations should be considered. Respondent validation, that is participants'revision of data collection and/or analysis [53], could have been used to increase the credibility of findings. However, this procedure was not carried out as it might burden the participants, who were expected to be focused on the initial stage of implementation of the SPLIT project. Additionally, the number of participants included may be seen as a potential limitation, particularly if this study's findings are considered in terms of empirical generalizability. It is important to take into account that the 7 seven health units, where the SPLIT project was going to be piloted, had at least one practitioner participating in the focus groups. Thus, the participants

could be considered representative of the practitioners that would be involved in the future implementation of the project. However, they cannot be considered representative of all GPs and PTs'perceptions regarding interdisciplinary stratified care for patients with LBP. These findings should be considered in terms of transferability, which is described as the ability to transfer the study's findings to individuals in contexts that may have some similarities to the context where the study was carried out [54]. Thus, some procedures were undertaken in order to enable the aforementioned transferability. For example, attention was given to describing the Portuguese context as well as the participants'characteristics and verbatim transcriptions from focus groups were presented to support the researchers'interpretations.

## Conclusions

This study provided the first insights into the perceptions of both GPs and PTs regarding the upcoming implementation of an interdisciplinary stratified model of care for patients with LBP. The use of qualitative research methods provided useful knowledge for the development and upcoming implementation of the SPLIT Project. The exploration of the acceptability, potential barriers and facilitators to the implementation of this project may provide an opportunity for future triangulation of data and, thus, for developing a deep understanding about the clinical outcomes and cost-efficacy as well as about the process of implementation and how they impact on each other. Further research focused on patients' perceptions regarding the interdisciplinary stratified model of care integrated into the SPLIT project is recommended.

## Acknowledgments

The authors would like to thank the General Practitioners and Physiotherapists for taking part in this study.

## Author Contributions

**Conceptualization:** Carmen Caeiro, Helena Canhão, Sofia Paiva, Luís A. Gomes, Rita Fernandes, Ana Maria Rodrigues, Rute Sousa, Fernando Pimentel-Santos, Jaime Branco, Ana Cristina Fryxell, Lília Vicente, Eduardo B. Cruz.

**Writing – original draft:** Carmen Caeiro, Helena Canhão, Eduardo B. Cruz.

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
