## [Decision Letter · Decision Letter 0]

25 Sep 2019

PONE-D-19-21570

Interdisciplinary stratified care for low back pain: a qualitative study on the acceptability, potential facilitators and barriers to implementation

PLOS ONE

Dear Dr. Caeiro,

Thank you for submitting your manuscript to PLOS ONE. After careful consideration, we feel that it has merit but does not fully meet PLOS ONE’s publication criteria as it currently stands. Therefore, we invite you to submit a revised version of the manuscript that addresses the points raised during the review process.

We would appreciate receiving your revised manuscript by Nov 09 2019 11:59PM. To enhance the reproducibility of your results, we recommend that if applicable you deposit your laboratory protocols in protocols.io, where a protocol can be assigned its own identifier (DOI) such that it can be cited independently in the future. For instructions see: http://journals.plos.org/plosone/s/submission-guidelines#loc-laboratory-protocols

We look forward to receiving your revised manuscript.

Kind regards,

Denis Bourgeois

Academic Editor

PLOS ONE

Journal Requirements:

Reviewers' comments:

Reviewer's Responses to Questions

**Comments to the Author**

1. Is the manuscript technically sound, and do the data support the conclusions?

Reviewer #1: Yes

2. Has the statistical analysis been performed appropriately and rigorously? 

Reviewer #1: N/A

3. Have the authors made all data underlying the findings in their manuscript fully available?

Reviewer #1: No

4. Is the manuscript presented in an intelligible fashion and written in standard English?

Reviewer #1: Yes

5. Review Comments to the Author

Reviewer #1: A well conducted and written qualitative study of two focus groups exploring the barriers and facilitators to implementing a stratified model of care for low back pain in Portugal primary care. The methods section in particular were very clearly written, and were easy to follow. The illustrative quotes used in the results provide good justification for the themes and sub-themes.

Suggestions:

1. I would have liked to have seen the use of some reporting guidelines, e.g http://www.equator-network.org/reporting-guidelines/coreq/

2. In the Methods section of the abstract, mention who was in the focus groups and how many people participated.

3. In the Conclusion section of the abstract, the final sentence is not a conclusion of this particular study. Delete and replace with a sentence that better sums up the findings of this particular study. In particular, this study was of only two focus groups, so the potential generalisability of the findings to the larger population of relevant practitioners is limited, so it would be helpful to mention that here.

4. P4, lines 77-81. Give more information here about the context for where this project is being implemented, e.g population demographics, number of GPs/PTs in this setting, etc. I did not get a sense for whether this is planned to be implemented in only one specific healthcare setting, or a larger setting. Please provide information about this.

5. P6, lines 124/5: typo “contribute to in financing health care…”. Reword

6. P7, line 152. Why did you decide that focus groups was the appropriate method for your research question? Why not individual interviews?

7. Table 1: probably best if this information is presented in the text as summary data, rather than individual data to reduce risk of potentially identifying individual participants.

8. P8, line 164: typo “synthetize”

9. P29. Line 648: provide some context here by relating the findings back to the SPLIT practitioners – were the participants at least representative of these practitioners?

6. PLOS authors have the option to publish the peer review history of their article (what does this mean?). If published, this will include your full peer review and any attached files.

Reviewer #1: Yes: Simon French

---

## [Author Response · Author response to Decision Letter 0]

23 Oct 2019

Response to “Review comments to the author”

Reviewer #1: A well conducted and written qualitative study of two focus groups exploring the barriers and facilitators to implementing a stratified model of care for low back pain in Portugal primary care. The methods section in particular were very clearly written, and were easy to follow. The illustrative quotes used in the results provide good justification for the themes and sub-themes.

Authors` comment: Thank you for the feedback provided.

Suggestions:

1. I would have liked to have seen the use of some reporting guidelines, e.g http://www.equator-network.org/reporting-guidelines/coreq/

Authors` response: Thank you for the suggestion. 

The consolidated Criteria for Reporting Qualitative Research (COREQ) had been followed. Therefore, the following sentence was introduced in the materials and methods section:

“The criteria for reporting qualitative research (COREQ) provided important guidelines for this study design as well as data analysis (28).” 

Additionally the following information was added in the reference list:

“28. Tong A, Sainsbury P, Craig J. Consolidated criteria for reporting qualitative research (COREQ): a 32-item checklist for interviews and focus groups. International Journal for Quality in Health Care. 2007;19: 349–357.”

2. In the Methods section of the abstract, mention who was in the focus groups and how many people participated.

Authors` response: Thank you for the suggestion. The following information was added:

“A qualitative study was carried out supported by two focus groups, which were conducted by two researchers. A focus group was carried out with each professional group. One focus group included six general practitioners and the other included six physiotherapists.” 

3. In the Conclusion section of the abstract, the final sentence is not a conclusion of this particular study. Delete and replace with a sentence that better sums up the findings of this particular study. In particular, this study was of only two focus groups, so the potential generalisability of the findings to the larger population of relevant practitioners is limited, so it would be helpful to mention that here.

Authors` response: Thank you for the suggestion. The information in the conclusion section of the abstract was replaced by the following sentences:

“This study`s participants highlighted the feasibility and acceptability of the SPLIT project. The identification of potential barriers and facilitators to its implementation also attained major relevance to better prepare the future implementation of this project. The generalizability of findings to the larger population of relevant practitioners is limited, since only two focus groups were carried out. Therefore, this study`s findings should be considered in terms of transferability to contexts that may have some similarities to the context where the study was carried out.”

4. P4, lines 77-81. Give more information here about the context for where this project is being implemented, e.g population demographics, number of GPs/PTs in this setting, etc. I did not get a sense for whether this is planned to be implemented in only one specific healthcare setting, or a larger setting. Please provide information about this.

Authors` response: Thank you for this feedback.

The authors considered information about the context fundamental to enable the transferability of this study`s findings to similar contexts. In order to provide detailed information about the context, a specific section entitled “context” had been introduced in the methods in order to avoid an extensive introduction. The authors had proposed this structure, since it has been used in some qualitative studies published by PLOS | One.

We consider that it would be useful to maintain the length of the introduction, so the reader won’t get lost in the rationale for this study. But, in agreement with the reviewer`s comment, we recognize that more information would be useful to provide a better sense of the context. Taking into consideration the reviewer`s suggestion, the following information was introduced in the “context section”:

“The SPLIT project was planned in partnership with one of the aforementioned administration bodies (ARS Lisbon and Tagus Valley) to be implemented in 1 Group of Health Centers (ACES Arrábida). This group of health centers provides health services to a population of 233.516 individuals, living in the municipalities of Palmela, Setúbal and Sesimbra (38). From the 22 health units included in this group of health centers, 7 were selected for piloting the project, based on the proportion of geographical area covered by ACES Arrábida (1 health unit from Palmela, 2 health units from Sesimbra and 4 health units in Setúbal). A total of 52 GPs and 3 PTs works in these 7 health units (38).”

The introduction of the aforementioned information may not be clear since there is a difference on the number of physiotherapists (3) working in the 7 health units where the project was going to be implemented and the number of physiotherapists who participated in the focus group (6). Therefore, the following information was added in the “participants” section:

“As there were only 3 PTs working in the health units where the SPLIT project was going to be implemented, the participation of more PTs from similar units was considered”.

The following information was added in the reference list:

“38. ARSLVT Núcleo de Estudos e Planeamento. ACES Arrábida, Caracterização e Propostas de Reestruturação [ACES Arrábida, Characterization and Proposals for Reorganization]. 2015 pp. 1–15.”

5. P6, lines 124/5: typo “contribute to in financing health care…”. Reword

Authors` response: Thank you. The sentence was rephrased as follows: 

“All residents have access to the health care provided by the NHS and participate in health care financing via co-payments and co-insurance”

6. P7, line 152. Why did you decide that focus groups was the appropriate method for your research question? Why not individual interviews?

Authors` response: Thank you. The justification for the use of focus groups was added in the “data collection section”:

“The focus group is defined as a discussion-based interview that produces data and insights via group interaction, which would be less accessible without the interaction founded in a group (41, 42). The focus groups were selected as a method for data collection since the aim of this study was focused on understanding the combined local perspectives of each professional group participating in the implementation of the SPLIT project. The research team aimed to explore potential agreements or disagreements as the discussion progressed and to grasp each professional group perceptions regarding the future implementation of the SPLIT project”.

Additionally, the following reference was added to the final list:

“41. Flick U. Focus groups. In: An introduction to qualitative research. 4th ed. Sage: 2009. pp. 194-209.”

7. Table 1: probably best if this information is presented in the text as summary data, rather than individual data to reduce risk of potentially identifying individual participants.

Authors` response: Thank you for this suggestion. Table 1 was removed from the manuscript. Following an example from a qualitative study published by PLOS | One, information about the participants` socio-demographic characteristics and experience was introduced in beginning of the findings section.

“Twelve participants (9 female, 3 male) were included. Five of the participants were 30-39, four were 40-49, and three were 20-29, 50-59 and 60-69, respectively. The GPs` group had an average of 13.17 years of professional experience, from which 10.17 years were based in primary care. The PTs` had an average of 21 years of professional experience, from which 13.87 years were based in primary care”.

8. P8, line 164: typo “synthetize”

Authors` response: Thank you. The word “synthetize” was replaced by “summarize”, as the latter was considered more adequate.

9. P29. Line 648: provide some context here by relating the findings back to the SPLIT practitioners – were the participants at least representative of these practitioners?

Authors` response: Thank you for this suggestion. The following information was added:

“It is important to take into account that the 7 seven health units, where the SPLIT project was going to be piloted, had at least one practitioner participating in the focus groups. Thus, the participants could be considered representative of the practitioners that would be involved in the future implementation of the project. However, they cannot be  considered representative of all GPs and PTs` perceptions regarding  interdisciplinary stratified care for patients with LBP.  These findings  should be considered in terms of transferability, which is described as the ability  to transfer the study`s findings to individuals in contexts that may have some  similarities to the context where the study was carried out”

---

## [Editor Report · Decision Letter 1]

4 Nov 2019

Interdisciplinary stratified care for low back pain: a qualitative study on the acceptability, potential facilitators and barriers to implementation

PONE-D-19-21570R1

Dear Dr. Caeiro,

We are pleased to inform you that your manuscript has been judged scientifically suitable for publication and will be formally accepted for publication once it complies with all outstanding technical requirements.

With kind regards,

Denis Bourgeois

Academic Editor

PLOS ONE
---

## [Editor Report · Acceptance letter]

8 Nov 2019

PONE-D-19-21570R1 

Interdisciplinary stratified care for low back pain: a qualitative study on the acceptability, potential facilitators and barriers to implementation 

Dear Dr. Caeiro:

I am pleased to inform you that your manuscript has been deemed suitable for publication in PLOS ONE. Congratulations! Your manuscript is now with our production department. 

With kind regards,

on behalf of

Professor Denis Bourgeois 

Academic Editor

PLOS ONE